# Concurrent training in cardiac rehabilitation: A scoping review of aerobic-strength combinations in patients with coronary artery disease

Aliff Latir[1,2]*, Eliza Hafiz[3], Anwar Suhaimi[1]*

**1** Department of Rehabilitation Medicine, Faculty of Medicine, Universiti Malaya, Kuala Lumpur, Malaysia, **2** Centre for Physiotherapy Studies, Faculty of Health Sciences, Universiti Teknologi MARA, Selangor Branch, Puncak Alam Campus, Bandar Puncak Alam, Selangor, Malaysia, **3** Faculty of Sports and Exercise Science, Universiti Malaya, Kuala Lumpur, Malaysia

☯ These authors contributed equally to this work.
\* aliff533@uitm.edu.my; anwars@um.edu.my

## Abstract

Concurrent training (CT), defined as the integration of aerobic and strength exercise modalities, is increasingly implemented within cardiac rehabilitation (CR) to improve cardiovascular and musculoskeletal health. However, CT prescriptions vary markedly, and the characteristics and application of CT within CR have not been comprehensively mapped. This scoping review examined how CT has been designed and delivered in CR and summarised reported outcomes across International Classification of Functioning, Disability and Health (ICF) domains. A systematic search of databases and supplementary sources was conducted from October 2023 to December 2025. Clinical trials evaluating CT compared with usual care or aerobic training alone were eligible. Fourteen trials (n = 1,037; 13 randomised, 1 single-group) were included. CT programmes varied widely in setting (hospital, community, hybrid), CR phase (II–III), duration (5–32 weeks), and training structure. Aerobic training was delivered using continuous aerobic training (CAT), aerobic interval training (AIT), or high-intensity interval training (HIIT), with intensities prescribed using peak oxygen uptake (VO$_2$peak), maximum heart rate (HR$_{max}$), heart rate reserve (HRR), work rate, ventilatory threshold, or rating of perceived exertion (RPE). Modalities ranged from cycle ergometers and treadmills to multimodal or circuit-based formats. Strength-training prescriptions differed in targeted muscle groups, sets (2–8), repetitions (8–20), equipment (e.g., machines, free weights, TheraBand), and intensity anchors (% of one repetition maximum, RPE, % of maximum voluntary contraction). Application of progressive overload was inconsistent across studies. Exercise capacity (primarily VO$_2$peak) and muscle strength were the most consistently assessed outcomes. No CT-related adverse events were reported. CT has been applied using diverse delivery formats and exercise prescriptions within CR. Physiological

**Data availability statement:** No new datasets were generated or analysed in this scoping review. All data supporting the findings of this study, including extracted study characteristics, synthesis tables and figures, the full electronic search strategies, and the PRISMA-ScR checklist, are provided within the manuscript and its Supporting Information files.

**Funding:** This review was supported by the Fundamental Research Grant Scheme (FRGS) from the Ministry of Higher Education Malaysia (Grant No. FRGS/1/2023/2210/UiTM/03/4, awarded to AL). The funder's website is available at: https://www.mohe.gov.my. The funder had no role in study design, data collection and analysis, decision to publish, or preparation of the manuscript.

**Competing interests:** The authors have declared that no competing interests exist.

outcomes were most consistently measured, whereas activity and participation-level outcomes showed greater variability and limited long-term evaluation. Future research should prioritise clearer reporting of training parameters, examine CT within community and hybrid CR models, incorporate behavioural and patient-reported outcomes, and investigate sex-specific responses and long-term effects to inform scalable and contextually adaptable CT approaches.

## Introduction

Cardiovascular diseases (CVD) remain a leading cause of morbidity and mortality worldwide, imposing a substantial burden on patients and healthcare systems [1,2]. Consequently, healthcare systems are managing a growing population of patients who require comprehensive long-term secondary prevention to reduce recurrent events, optimise functional recovery, and maintain health-related quality of life [3]. Cardiac rehabilitation (CR) is a cornerstone of secondary prevention. CR is a structured, multidisciplinary intervention that encompasses medical evaluation, risk factor modification, exercise training, psychosocial support, and patient education, all designed to enhance recovery, restore functional independence, and reduce the risk of future cardiovascular events [4]. Since the mid-20th century, CR has evolved from a model of passive recovery to a comprehensive, multidisciplinary programme incorporating physical exercise, nutritional guidance, and psychosocial support [1,5]. Contemporary CR prioritizes structured aerobic and resistance training as core components, both of which have been empirically demonstrated to improve functional capacity and reduce the risk of recurrent cardiac events [6,7]. The programme further enhances quality of life, reduces hospital readmissions, and lowers mortality rates [8,9].

A central challenge in CR is the optimisation of exercise prescriptions, particularly the integration of aerobic and strength modalities to maximise rehabilitation outcomes [6,10]. While aerobic training is well established as a cornerstone of CR, the role and implementation of strength training, either alone or in combination, remain less clearly defined [11,12]. Knowledge gaps persist regarding the most effective exercise prescriptions and the health impact of concurrent training (CT) [1,2]. The training is typically characterised as a regimen incorporating both aerobic and resistance training, delivered either within the same session or on a separate day throughout the training programme, in contrast to aerobic-only training [10,13,14]. Ongoing debates also address the potential interference effect between aerobic and strength training and its implications for muscular and cardiovascular adaptations [10,15].

The International Classification of Functioning, Disability and Health (ICF) framework provides an important perspective for evaluating the effects of CT [16]. Unlike approaches that focus solely on physiological endpoints, the ICF supports a multi-dimensional classification of health outcomes, capturing body functions, activities, participation, and contextual factors [17]. ICF-based patient-reported outcome measures (PROMs) and registries have strengthened the assessment of functional status

and health-related quality of life, supporting benchmarking and continuous quality improvement in CR [18–21]. However, many existing assessment instruments used in CR focus predominantly on body functions, with limited consideration of activity-level, environmental, or personal factors within the ICF framework [22]. Incorporating the ICF, therefore, promotes a more comprehensive evaluation of CT, reflecting not only physiological improvements but also meaningful changes in functional recovery and daily living.

Given these uncertainties, interest in CT continues to grow. However, heterogeneity in programme delivery, exercise characteristics, and outcome measures has hindered consensus on best practice. A comprehensive mapping of existing evidence is therefore needed to clarify how CT has been implemented in CR, describe the characteristics of exercise prescriptions and ICF-aligned outcome domains, and identify gaps that limit comparability across studies. This scoping review addresses these objectives by characterising CT protocols and summarising the breadth of available evidence to inform future research and clinical practice.

## Methodology

### Design

This scoping review follows a standardised methodological approach [23] and is officially registered on the Open Science Framework (https://osf.io/naubh). Selecting and screening articles for inclusion followed the criteria of the Preferred Reporting Items for Systematic Reviews and Meta-Analyses extension for Scoping Reviews (PRISMA-ScR) (Fig 1) [24].

### Eligibility criteria

Using the PCC framework [25], the authors assessed and included all relevant articles. **Population (P):** patients with coronary artery disease (CAD) across all genders and ethnic backgrounds. **Concept (C):** CT protocols integrating aerobic and strength modalities, with outcomes evaluated across ICF domains, including body functions, activities, and participation. For this review, aerobic interval training (AIT) was defined as structured interval-based aerobic exercise performed at moderate-to-high intensities [26], whereas high-intensity interval training (HIIT) was defined as interval-based aerobic exercise prescribed at vigorous-to-near-maximal intensities [27]. These definitions were used to guide study classification during screening and data synthesis. **Context (C):** Cardiac rehabilitation delivered in hospital, community, or home-based settings, without restriction on geographical location or healthcare system. To ensure comprehensive mapping of CT protocols, all empirical study designs, including randomised controlled trials, non-randomised trials, pilot and, feasibility studies and quasi-experimental studies were eligible for inclusion. Exclusion criteria were as follows: (i) studies lacking a full-text description; (ii) non-English language; and (iii) case reports, reviews, conference abstracts without sufficient details or correspondence.

### Search strategy

A comprehensive search was performed across four databases (Scopus, Cochrane Library, Web of Science, and PubMed) from October 2023 to March 2025, encompassing all studies published or indexed within this period. The search was updated in December 2025 to capture recently published or indexed articles, ensuring no recent evidence relevant to CT was overlooked prior to finalising the review. The review considered studies published between 2000 and 2025, reflecting contemporary developments in CT protocols. The search employed the specified keywords and Boolean operators: ("coronary artery disease" OR "ischemic heart disease" OR "myocardial infarction" OR "CAD") AND ("cardiac rehabilitation" OR "exercise therapy" OR "rehabilitation") AND ("resistance training" OR "strength training" OR "weight training" OR "resistance exercise") AND ("aerobic training" OR "endurance training" OR "aerobic exercise" OR "cardiorespiratory fitness"). To ensure comprehensive coverage, supplementary sources were searched, including Google Scholar, institutional repositories, and manual reference checking of relevant articles. The full search strategy for each database is provided in Supplementary File 1.

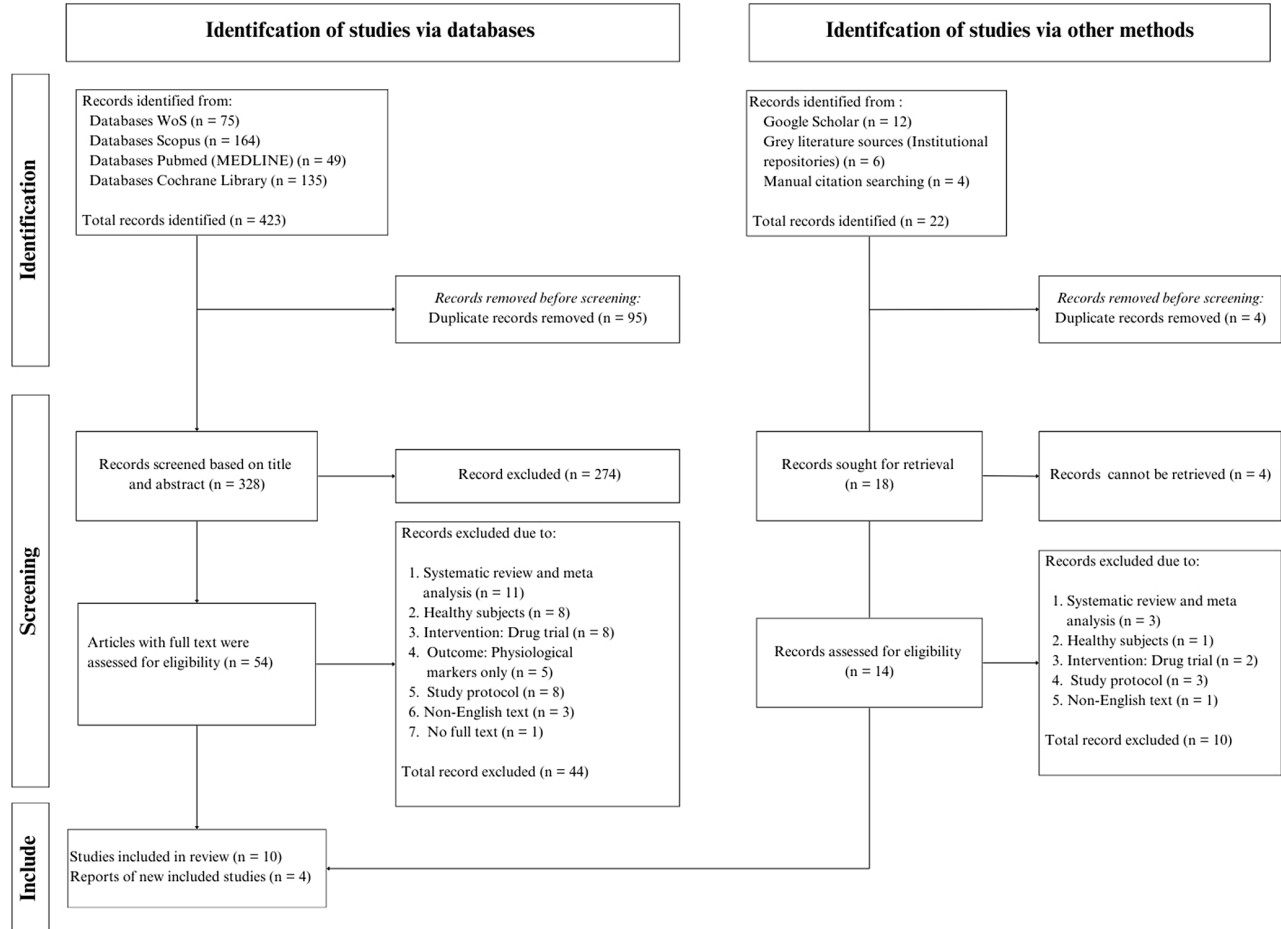

**Fig 1. PRISMA-ScR flow diagram of study selection.**

## Study selection and data extraction

Three reviewers (A.L., A.S., and E.H.) independently screened all citations and full-text articles identified through the literature search to determine eligibility. Any disagreements were resolved through consensus. Data extraction was performed autonomously by each reviewer using a standardised approach. Author A.L. completed the initial extraction using End-Note software, and all three reviewers conducted duplicate extractions to ensure accuracy. The extracted data included (i) study characteristics (authors, study design, country of origin and publication year), (ii) participants details (age, gender and sample size), (iii) exercise characteristics of CT and associated health-related outcome measures. All authors confirmed and approved the final dataset.

## Evaluation of reporting quality

Although optional in scoping reviews, reporting quality was assessed to characterise the existing CT literature. The methodological quality of each included study was assessed using the Rosendal Scale [28], which evaluates factors that may introduce bias related to participant selection, performance, and data analysis. The scale integrates elements from the CONSORT statement [29], the Delphi List [30], the Physiotherapy Evidence Database (PEDro) scale [31], and the

Jadad scoring system [32]. A score of ≥ 60% indicates good methodological quality [28]. The score (percentage) for each study was calculated by dividing the number of 'yes' responses by the 15 applicable items in the scale. All studies were assessed independently by all authors, and any discrepancies were resolved through discussion.

### Synthesis of data

A narrative synthesis approach was used to map the characteristics of CT protocols across studies. Aerobic and strength components were categorised according to the FITT principles (frequency, intensity, time, and type), and outcome measures were grouped into ICF domains (body structure and function, activity levels, and participation). Tables were developed to summarise protocol characteristics and outcome domains across the included study designs.

## Results

### Study selection

A systematic search of four databases identified 423 records. After removing duplicates, 328 records remained for screening, of which 274 excluded at the title/abstract stage. Fifty-four texts were assessed, and 44 were excluded for not meeting eligibility criteria, resulting in ten studies included from database searches. Additional sources identified through Google Scholar, grey-literature, and manual reference checking contributed 22 records. After removing duplicates, 18 records were sought for retrieval. Four records were unretrievable, leaving fourteen records screened for eligibility. Four studies met the inclusion criteria. Combining database and supplementary sources, a total of 14 studies, published between 2000–2025, were included in this review [13,26,33–44].

### Reporting quality of included studies

Results of the quality assessment are presented in Table 1. Average Rosendal score was 61.2%, with nine achieving a score rated as good quality (≥ 60%).

### Characteristics of included studies

Thirteen of included studies were randomised controlled trials [13,26,33–42,44], and one employed a single-group repeated-measures design [43]. By country, three studies were conducted in the USA, two in Canada, and one each in Slovenia, Finland, Germany, Israel, Belgium, Iran, France, Japan, and Greece. The total sample size across all included studies was 1,037 participants (range: 16–391). The mean age of participants ranged from 52.3 to 69.2 years, consisting predominantly of males (67%, n=626). Most studies (n=9) included both genders [13,26,34,36–38,41–43], while three studies recruited only males [33,39,40] and two studies included only females [35,44]. Two studies did not report mean participant age [44] or provided only an age range [40]. All studies recruited patients with stable CAD, characterised by acute myocardial infarction (MI), stenosis of at least one major coronary artery, stable angina pectoris, post-percutaneous coronary intervention (PCI), or post-coronary artery bypass grafting (CABG). Three studies applied an inclusion criterion of left ventricular ejection fraction (EF) > 45%, whereas one study included participants with EF < 45% [40].

### Context of CT delivery

Eleven studies implemented CT in hospital-based CR programmes [13,26,33,34,37–42,44], two study delivered CT in community-based settings [35,36], and one study employed hybrid-based model [43]. The phase of CR also varied between studies: Seven were conducted in Phase II [13,37–40,42,44], five in Phase III [26,33,34,36,43], and two did not specify the CR phase [35,41]. The mean duration of the CR programmes was 16.2±8.9 weeks (range 5–32 weeks), with exercise frequency ranging from one to five sessions per week. Ten studies implemented CT by integrating aerobic and strength training within a single session [13,26,34–36,38–41,43]. Three studies adopted a phased approach, commencing

**Table 1. Methodology quality assessment summary and Rosendal score of studies included in this scoping review.**

| Author | Eligibility[1] | Randomisation[2] | Method of Randomisation[3] | Sample size justified[4] | Pre-trial conditions[5] | Baseline measures[6] | Blinding of subjects[7] | Blinding of investigators[8] | Method and Evaluation of Blinding[9] | Non-Completers Described[10] | Statistics described[11] | Measures and Variability Described[12] | Between Group Stats Comparisons[13] | Method used to access adverse effects was described[14] | Appropriate washout period[15] | % Score^ |
|---|---|---|---|---|---|---|---|---|---|---|---|---|---|---|---|---|
| Moghadam et al. [34] | 1 | 1 | 1 | 0 | 0 | 1 | 1 | 1 | 0 | 1 | 1 | 1 | 1 | 1 | NA | 78.6 |
| Lehti et al. [26] | 1 | 1 | 1 | 0 | 0 | 1 | 0 | 1 | 0 | 1 | 1 | 1 | 0 | | NA | 64.3 |
| Kambic et al. [13] | 1 | 1 | 1 | 1 | 0 | 1 | 0 | 0 | NA | 1 | 1 | 1 | 1 | 1 | NA | 76.9 |
| Izawa et al. [36] | 1 | 1 | 0 | 0 | 0 | 1 | 0 | 0 | NA | 1 | 1 | 1 | 0 | 0 | NA | 46.2 |
| Hussein et al. [37] | 1 | 1 | 1 | 0 | 0 | 1 | 0 | 0 | NA | 1 | 1 | 1 | 0 | 1 | NA | 61.5 |
| Hansen et al. [38] | 1 | 1 | 1 | 0 | 1 | 1 | 0 | 0 | NA | 1 | 1 | 1 | 0 | 0 | NA | 61.5 |
| Gayda et al. [39] | 1 | 1 | 0 | 0 | 0 | 1 | 0 | 0 | NA | 0 | 1 | 1 | 0 | 0 | NA | 38.5 |
| Dor-Haim et al. [40] | 1 | 1 | 1 | 0 | 0 | 1 | 0 | 0 | NA | 1 | 1 | 1 | 1 | 1 | NA | 69.2 |
| Deka et al. [41] | 1 | 1 | 1 | 1 | 0 | 1 | 0 | 1 | 0 | 1 | 1 | 1 | 1 | 1 | NA | 78.6 |
| Currie et al. [42] | 1 | 1 | 0 | 0 | 0 | 1 | 0 | 0 | NA | 1 | 1 | 1 | 0 | 0 | NA | 46.2 |
| Christle et al. [43] | 1 | NA | NA | 0 | 0 | 1 | 0 | 0 | NA | 1 | 1 | 1 | NA | 1 | NA | 60.0 |
| Arthur et al. [44] | 1 | 1 | 1 | 1 | 0 | 1 | 0 | 1 | 0 | 1 | 1 | 1 | 0 | 1 | NA | 71.4 |
| Volaklis et al. [33] | 1 | 1 | 0 | 0 | 0 | 1 | 0 | 0 | NA | 1 | 1 | 1 | 0 | 1 | NA | 50.0 |
| Khadanga et al. [35] | 1 | 1 | 0 | 1 | 0 | 1 | 0 | 0 | NA | 1 | 1 | 1 | 0 | 0 | NA | 53.8 |

Abbreviation: NA, not available

with aerobic training before integrating strength training in later phases [37,42,44]. One study delivered aerobic and strength training in separate sessions, with participants completing four sessions per week divided equally between aerobic and circuit-based resistance training [33].

## Exercise characteristics of CT protocols

**Aerobic training characteristics.** A summary of the characteristics of CT protocols is presented in Table 2. The type and duration of aerobic training varied across studies. Six studies employed 20–50 minutes of continuous aerobic training (CAT) [33,34,36,38,39,42], five used 20–45 minutes of AIT [13,26,37,40,43], and three implemented HIIT [35,41,42]. Among the included studies, six used single exercise modality [13,26,35,36,41,42], six used multi-exercise modalities [33,34,37–39,44] and one used a circuit-based approach [40]. One study did not provide a report on the exercise modalities [43]. Exercise modalities included cycle ergometers, treadmills, arm ergometers, stair climbers, NuStep machines, rowers, and stair masters.

The exercise intensity prescribed was based on the types of aerobic training. For CAT, intensity was set at: (i) 51–65% of pre-training peak work rate ($W_{peak}$), which represents maximum physical exertion achieved during cardiopulmonary exercise testing (CPET) [42]; (ii) 65% of peak oxygen uptake ($\dot{V}O_2peak$) [38]; (iii) 60–90% of maximum heart rate ($HR_{max}$) [33,34]; (iv) 11–13 on Borg's rating of perceived exertion (RPE) scale [36]; or (v) an individualised workload based on ventilatory threshold (VT) and power (W), whereby training power (W) was prescribed relative to the power output achieved at VT [39]. For AIT, reported intensity included: (i) 50–80% of $W_{peak}$ [13]; (ii) 50–85% of heart rate reserve (HRR) [26,40]; (iii) target heart rate ($HR_{target}$) prescribed using HRR (Karvonen formula), calculated as 40–60% of the difference between $HR_{max}$ achieved during the stress test and resting heart rate, added to resting heart rate [37]; (iv) 40–70% of $\dot{V}O_2peak$ [44]; and (v) a combination of $HR_{max}$ and Borg's RPE 11–13 in hybrid CT [43]. Studies employing HIIT prescribed alternating bouts of high- and low- intensity aerobic, such as: (i) 85–90% $HR_{max}$ with recovery at 60–70% $HR_{max}$ [41], (ii) 90–95% $HR_{max}$ with recovery at 50–60% $HR_{max}$ [35], or (iii) 85% $W_{peak}$ with recovery at 10% $W_{peak}$ [42]. In terms of progressive exercise loading, four studies gradually increase the exercise intensity [13,26,33,44], and three studies adjusted exercise intensity and duration inversely, whereby increases in intensity were accompanied by corresponding reductions in exercise duration [34,42,43]. Seven studies maintained a constant exercise intensity throughout the programme [35–41].

**Strength training characteristics.** The frequency and volume of strength training ranged from 2 to 8 sets of 8–20 repetitions per session. Exercise modalities included body weight exercises, dumbbells, TheraBands, leg press machines, Cybex machines, and specialised strength apparatus. Of the included studies, eleven adopted multi-exercise format; four targeted major muscle groups of both the upper and lower limbs [26,40–42] and five focused primarily on lower-limb musculature [36,37,39,43,44]. Two studies used single-exercise format targeting the lower limbs [13,35] and two incorporated circuit training approach [33,34].

The prescribed intensity of strength training varied across studies. Nine studies prescribe 30–80% of one repetition maximum (1RM) [13,26,33,35,37,38,40,41,44], while one study used 40–80% of the two repetitive maximum (2RM). Three studies applied Borg's RPE scale at 11–15 [36,42,43], and two studies prescribed 40% maximal voluntary contraction (MVC) [34,39]. Progressive overload was incorporated in seven studies, achieved through gradual increases in intensity [40,44], incremental increases in resistance load [33,34,42] or a combination of parameters such as intensity, repetitions or volume [13,43].

## Outcome domains mapped to the ICF framework

Across studies, outcomes were reported at the level of body function and structure, activity and participation. A structured summary of these measures, mapped to the ICF framework, is provided in Table 3.

**Table 2. Characteristics of CT studies.**

| Author (Country) | Study design | Participant characteristics | | | Training characteristics | | | | Mode of Delivery (CR Phase) |
|---|---|---|---|---|---|---|---|---|---|
| | | Diagnosis | Age (yr) | Sex (M/F) | Group (n) | AT Duration/ Type/ Modalities/ Intensity | ST Sets/ repetitions/ no. of exercises/ %RM | Duration (weeks) Exercise frequency (times/ week) | |
| Deka et al. [41] (USA) | RCT | CAD, preserved EF | 69.2±4.9 | 68/22 | UC (45) | Conventional medical treatment | NA | NA | Supervised (NA) |
| | | | | | CT (45) | 20 min HIIT, Treadmill walking, 85-90% $HR_{max}$/ 60–70 $HR_{max}$ | 2 sets 15 reps 10 exercises 30-50% 1RM | 8 1x AT 1x ST | |
| Kambic et al. [13] (Slovenia) | RCT | CAD, ACS and/ PCI, preserved EF | 61.0±8.0 | 44/15 | AT | 45 min Aerobic Interval, Cycling, 50→80% $W_{peak}$ | NA | 12 3x AT | Supervised (Phase II) |
| | | | | | CT (HL-ST) | 45 min Aerobic Interval, Cycling, 50→80% $W_{peak}$ | 3 sets 6-8 reps 1 exercise 70%→80%1RM | 12 3x AT 3x ST | |
| | | | | | CT (LL-ST) | 45 min Aerobic Interval, Cycling, 50→80% $W_{peak}$ | 3 sets 12-22 reps 1 exercise 35%→45%1RM | 12 3x AT 3x ST | |
| Lehti et al. [26] (Finland) | RCT | Stable CAD, coronary artery stenosis | 61.0±7.3 | 20/3 | C (11) | Usual daily activities | NA | NA | Supervised (Phase III) |
| | | | | | CT (12) | 20 min Aerobic Interval Cycle ergometer, 50 - 60% HRR→85% HRR | 2-3 sets 12 reps 5 exercises 50-70% 1RM | 21 2x AT 2x ST | |
| Christle et al. [43] (Germany) | SGRM | CAD, 1 vessel disease | 63.75±10.25 | 326/65 | CT (391) | 27-33 min Aerobic interval, Borg RPE 11–13 | 2-3 sets 15-20 reps 4 exercises Borg RPE 11–13 | 24 4x AT 4x ST | Hybrid (Phase III) |
| Dor-Haim et al. [40] (Israel) | RCT | CAD, acute MI with EF<45% | 47 - 69 | 48/0 | AT (26) | 45 min Aerobic training, Cycling, 60 −70% HRR | NA | 12 2x AT | Supervised (Phase II) |
| | | | | | CT (SCT) (22) | 24 min Aerobic Interval, Cycling, 75 - 85% HRR | 8 sets 15 reps 8 exercises 30%→50% 1RM | 12 2x AT 2x ST | |
| Hussein et al. [37] (USA) | RCT | CAD, <6months, MI/ CABG | 60.4±13.3 | 25/25 | AT (25) | 20 min Aerobic Interval, Bike, treadmill, Nu-step machine, rower, stair master, arm-leg, HR target (40–60% heart rate range + resting heart rate) | NA | 12 3x AT | Supervised (Phase II) |
| | | | | | CT (25) | 20 min Aerobic Interval, Bike, treadmill, Nu-step machine, rower, stair master, arm-leg, HR target (40–60% heart rate range + resting heart rate) | 2 sets 8-12 reps 3 exercises 60% 1RM | 12 3x AT 3x ST | |

*(Continued)*

**Table 2.** (Continued)

| Author (Country) | Study design | Participant characteristics | | | Training characteristics | | | | Mode of Delivery (CR Phase) |
|---|---|---|---|---|---|---|---|---|---|
| | | Diagnosis | Age (yr) | Sex (M/F) | Group (n) | AT Duration/ Type/ Modalities/ Intensity | ST Sets/ repetitions/ no. of exercises/ %RM | Duration (weeks) Exercise frequency (times/ week) | |
| Currie et al. [42] (Canada) | RCT | CAD | 66±8 | 18/1 | CT (MIT+ST) (11) | 30→50 min MIT, Cycling, 51-65% $W_{peak}$ | 2 sets 10-12 reps 7 exercises Borg's RPE score 11-15 | 24 2x MIT 2x ST | Supervised Phase II |
| | | | | | CT (HIIT+ST) (9) | 10 min HIIT, Cycling, 85% $W_{peak}$/ 10% $W_{peak}$ | 2 sets 10-12 reps 7 exercises Borg's RPE score 11-15 | 24 2x HIIT 2x ST | |
| Hansen et al. [38] (Belgium) | RCT | CAD | 59.7±8.1 | 57/3 | AT (30) | 40 min AT, Cycling, Walking, Arm Cranking, 65% $VO_2$peak | NA | 6 3x AT | Supervised Phase II |
| | | | | | CT (30) | 40 min AT, Cycling, Walking, Arm Cranking, 65% $VO_2$peak | 2 sets 12–20 reps 2 exercises 65% 1RM | 6 3x AT 3x ST | |
| Gayda et al. [39] (France) | RCT | CAD | 55±8 | 16/0 | AT (8) | 30 min AT, Cycle ergometer, individualized based on VT and W+60 min walk | NA | 7 3x AT | Supervised Phase II |
| | | | | | CT (8) | 30 min AT, Cycle ergometer, individualized based on VT and W | 3 sets 10 reps 2 exercises 40% MVC | 7 3x AT 3x ST | |
| Moghadam et al. [34] (Iran) | RCT | CAD, post CABG | 52.3±5.9 | 60/28 | AT (22) | 30 min AT, Treadmill, cycle ergometer, arm crank 60% $HR_{max}$ | NA | 5 5x AT | Supervised (Phase III) |
| | | | | | CT (40-ST) (22) | 30 min AT, Treadmill, cycle ergometer, arm crank 60% $HR_{max}$ | 2 sets 12 reps 5 exercises 40% 2RM | 5 5x AT 5x ST | |
| | | | | | CT (60-ST) (22) | 30 min AT, Treadmill, cycle ergometer, arm crank 60% $HR_{max}$ | 2 sets 12 reps 5 exercises 60% 2RM | 5 5x AT 5x ST | |
| | | | | | CT (80-ST) (22) | 30 min AT, Treadmill, cycle ergometer, arm crank 60% $HR_{max}$ | 2 sets 12 reps 5 exercises 80% 2RM | 5 5x AT 5x ST | |

| Author (Country) | Study design | Participant characteristics | | | Training characteristics | | | | Mode of Delivery (CR Phase) |
|---|---|---|---|---|---|---|---|---|---|
| | | Diagnosis | Age (yr) | Sex (M/F) | Group (n) | AT Duration/ Type/ Modalities/ Intensity | ST Sets/ repetitions/ no. of exercises/ %RM | | Duration (weeks) Exercise frequency (times/ week) | |
| Arthur et al. [44] (Canada) | RCT | CAD | NA | 0/92 | AT (46) | 40 min AT, Cycle ergometer, treadmill, arm ergometer, stair climber, 40%−70% VO$_2$peak | NA | | 24 2x AT | Supervised (Phase II) |
| | | | | | CT (46) | 40 min AT, Cycle ergometer, treadmill, arm ergometer, stair climber, 40%−70% VO$_2$peak | UE, 2 sets 8-10 reps 2 exercises 50%→70% 1RM | LE, 2 sets 10-12 reps 2 exercises 30%→70% 1RM | 24 2x AT 2x ST | |
| Izawa et al. [36] (Japan) | RCT | CAD | 66.3 | 16/2 | AT (8) | 60 min AT, Walking, 11-13 Borg's RPE | NA | | 24 2x | Unsupervised (Phase III) |
| | | | | | CT (10) | 30 min AT, Walking, 11-13 Borg's RPE | 4 sets 5 reps 2 exercises 11-13 Borg scale | | 24 2x AT 2x ST | |
| Volaklis et al. [33] (Greece) | RCT | CAD | 56 ± 10 | 27/0 | C (13) | No exercise | NA | | NA | Supervised (Phase III) |
| | | | | | CT (14) | 20 min AT, Walking, Running, Cycling, 60-75% HR$_{max}$→70–85% HR$_{max}$ | 3 sets 12-15 reps 8 exercises 60% 1RM | | 32 4x AT 4x ST | |
| Khadanga et al. [35] (USA) | RCT | CAD | 65.2 ± 11.1 | 0/56 | AT (27) | 45 min MCT Treadmill walking, Ergometer 70-85% HR$_{max}$ | NA | | 12 3x MCT | Supervised (Phase III) |
| | | | | | CT (HIIT+HL-ST) (29) | HIIT Treadmill walking, 5 min warmup 50–60% HR$_{max}$ 4-minute intervals at 90%−95% of HR$_{max}$ 4-minute recovery period at 50%−60% of HR$_{max}$ | 2 sets 10 reps 1 exercise 80% 1RM | | 12 3x HIIT 3x ST | |

ACS, acute coronary syndrome; AT, aerobic training; Borg's RPE, borg's rating of perceived exertion; C, control group; CABG, coronary artery bypass grafting; CAD, coronary artery disease; CT, concurrent aerobic-strength training; EF, ejection fraction; HR$_{max}$, maximum heart rate; HRR, heart rate reserve; HIIT, high intensity interval training; HL, high load; LL, low load; LE, lower extremities; MI, myocardial infarction; MCT, moderate continuous training; MIT, moderate intensity training; MVC, maximum voluntary contraction; NA, Not available; PCI, percutaneous coronary intervention; RCT, randomized control trial; SCT, super circuit training; SGRM, single group repeated measure design; ST, strength training; UC, usual care; UE, upper extremities; VO$_2$, oxygen uptake; VT, ventilatory threshold; W, power; W$_{peak}$, peak work rate; 1RM, 1-repetitive maximum; 2RM, 2-repetitive maximum; →, progressive load

**Table 3. Overview of outcome measures reported in CT studies mapped to ICF domains.**

| Author | Outcome measures | | | | | Main findings | AE/Hosp |
|---|---|---|---|---|---|---|---|
| | Body function and structure | | Activities | | Participation | | |
| | Exercise capacity | Muscle Strength | Functional capacity | Physical Activity | Health-related quality of Life | | |
| Deka et al. [41] | NA | NA | ISWT | IPAQ | SF-36 | • Significant group and time interaction vs. UC for BMI ($p<0.001$), body fat (%) ($p<0.001$), and waist circumference ($p<0.001$), functional performance ($p<0.001$), PA ($p<0.001$) and HrQoL ($P<0.001$). | No adverse events |
| Kambic et al. [13] | CPET | LE: 1RM prediction equations UE: hand grip dynamometer, arm curl test | 6MWT | NA | NA | • All groups had no significant impact on body composition measures.<br>• There was a significant time x group interaction for the 6MWT.<br>• HL-ST statistically significant improved gait speed, arm curl, and TUG compared with the AT group, and LL-ST showed greater improvement TUG compared with AT.<br>• There were no differences between HL-ST and LL-ST in post-training improvement in any of the physical performance measures. | NA |
| Lehti et al. [26] | Exercise Stress Test | Chair dynamometer | NA | NA | NA | • CT has insignificant improvement in all body composition measures.<br>• VO$_2$peak (6.9%, $p<0.05$), exercise time (11.2%, $p<0.05$) and muscle strength (19.9%, $p<0.05$) improved significantly after 12 weeks of training in the CT compared to C. | No adverse events |
| Christle et al. [43] | CPET | NA | NA | NA | NA | • Abdominal girth reduced significantly ($p<0.001$)<br>• VO$_2$peak and W$_{peak}$ increased significantly, 21.8±6.1 to 22.8±6.3 mL/kg/min and 128±39–138±43 W, respectively (both $p<0.001$, respectively). | No adverse events |
| Dor-Haim et al. [40] | Exercise Stress Test (Bruce Protocol) | Hand Grip Strength | NA | NA | SF-12 | • MET of the CT group was significantly higher than the AT group ($p=0.008$).<br>• CT produce significant improvements in muscle strength comparable to CAT.<br>• There was no significant between group differences in HrQoL scores. | No adverse events |
| Hussein et al. [37] | NA | 1RM | METs | NA | NA | • Percent body fat was reduced for CT after training ($p<0.01$) with significant difference in between groups ($p<0.01$).<br>• The relative gain in lean mass was greater in CT ($p=0.0006$).<br>• Strength gains for CT were greater than for group I on the three strength machines ($p<0.01$). | No adverse events |
| Currie et al. [42] | CPET | NA | NA | NA | SF-36 | • All body composition indices and HrQoL scores were unchanged with CT.<br>• VO$_2$peak increased from pretraining to 3-months in both groups (MIT+ST: 19.8±7.3 vs. 23.2±7.4 ml kg$^{-1}$ min$^{-1}$; HIIT+ST: 21.1±3.3 vs. 26.4±5.2 ml kg$^{-1}$ min$^{-1}$, $p<0.001$) with no further increase at 6-months. | NA |
| Hansen et al. [38] | CPET | Isokinetic dynamometer | NA | IPAQ | NA | • Maximal exercise capacity, ventilatory threshold increased and steady-state exercise respiratory exchange ratio reduced significantly.<br>• Muscle performance increased.<br>• Baseline habitual activity was not different between groups ($p=0.68$). | No adverse events |

*(Continued)*

**Table 3.** (Continued)

| Author | Outcome measures | | | | | Main findings | AE/Hosp |
|---|---|---|---|---|---|---|---|
| | **Body function and structure** | | **Activities** | | **Participation** | | |
| | Exercise capacity | Muscle Strength | Functional capacity | Physical Activity | Health-related quality of Life | | |
| Moghadam et al. [34] | NA | 2RM | MET | NA | NA | • CT groups showed significantly greater muscle strength than those in the AT group ($p \leq 0.05$).<br>• The net gains in MET of CT groups were significantly greater than that for the AT group ($p \leq 0.05$). | No adverse events |
| Gayda et al. [39] | CPET | Isokinetic dynamometer | NA | NA | NA | • Both CT and AT groups showed significant improvements in VO$_2$peak and W$_{peak}$.<br>• Skeletal muscle function data were not significantly modified after CT. | NA |
| Arthur et al. [44] | CPET | 1RM | NA | NA | SF-36 | • Both groups demonstrated similar significant improvements in VO$_2$peak (CT +19% vs AT +22%).<br>• There were no between-group differences in strength.<br>• PCS of HrQoL increased sig. in both groups, however the observed increase was not different between groups ($p = 0.52$). | No adverse events |
| Izawa et al. [36] | CPET | Isokinetic dynamometer | NA | Pedometer | NA | • VO$_2$peak was not significantly different from initial values within each group, and there were no statistically significant interaction periods by group.<br>• Knee muscular strength in the CT group was significantly higher than that in the AT group.<br>• No significant group and time interactions were detected for PA ($p = 0.09$), and there were no significant differences in pedometer values between CT and AT group. | NA |
| Volaklis et al. [33] | Bruce protocol | 1RM | NA | NA | NA | • Exercise time, VO$_2$peak, minute ventilation and respiratory quotient increased significantly after 4 months of CT.<br>• Muscular strength increased significantly in all exercises by an average of 14% (range 12–16%) after 4 months and by 29% (range 22–36%) after 8 months of training. | No adverse events |
| Khadanga et al. [35] | VO$_{2\ peak}$ | 1RM | NA | NA | NA | • VO$_2$peak increased more in the CT group (+23%) than in the AT group (+7%) ($p = 0.03$).<br>• CT group also showed greater improvement in leg strength compared to the AT ($p = 0.004$). | No adverse events |

AE, adverse events; AT, aerobic training; C, control group; CPET, cardiopulmonary exercise testing; CT, concurrent aerobic-strength training; HrQoL, health-related quality of life; Hosp, hospitalisation; HL-ST, high load strength training; IPAQ, International Physical Activity Questionnaire; ISWT, Incremental Shuttle Walking Test; LE, lower extremities; LL-ST, low load strength training; MET; metabolic equivalent of task; NA, Not available; PA, physical activity; PCS, physical component summary score; SF-12, 12-Item Short Form Health Survey; SF-36, 36-Item Short Form Health Survey; TUG, times up and go test; UC, usual care; UE, upper extremities; VO$_2$peak, peak oxygen uptake; W$_{peak}$, peak work rate; 1RM, 1-repetitive maximum; 2RM, 2-repetitive maximum; 6-MWT, 6-min walk test

**Body functions and structures.** Exercise capacity was assessed across studies using either CPET [13,35,36,38,39,42–44] or modified exercise stress-testing protocols [26,33,40]. The VO$_2$peak was the most frequently reported outcome. Most studies (n = 6) evaluating CT reported significant increases in VO$_2$peak from baseline [42,43] or greater improvements when compared with aerobic-only training [35,38] or control (i.e., no exercise prescription) [26,33]. However, one study found no additional improvement in VO$_2$peak with CT with relative to aerobic training alone [36], indicating variability in how CT influences aerobic capacity across training designs. Overall, the evidence indicates that

VO₂peak is a consistent measure of aerobic adaptation in CT protocols; although improvements are generally observed, the magnitude and comparative effects vary across studies.

Muscle strength outcomes were evaluated using a range of instruments, including one-repetition maximum (1RM) [13,33,35,37], two-repetition maximum (2RM) [34], chair dynamometry [26], hand-grip dynamometry [13,40], or isokinetic dynamometer [36,38,39]. Most studies reported improvements in strength over time in groups receiving CT, although the magnitude of change varied according to the testing modality, muscle group assessed, and prescribed training intensity [26,33–38,40]. A minority of studies reported comparable improvements between CT and aerobic group [44] or slightly higher isometric force output during isometric muscle testing [39], indicating heterogeneity in responsiveness across strength domains and CT formats. Overall, the included studies illustrate the diverse ways in which strength outcomes are operationalised and the range of responses observed following CT.

**Activity.** Functional performance was assessed using the incremental shuttle walk test (ISWT) [41], 6-minute walk test (6MWT) [13], or estimated metabolic equivalents (METs) [34,37]. Across these studies, CT was generally associated with improvements in functional capacity from baseline [13,41], although direct comparisons between protocols or variations in training intensity (e.g., differing strength-training loads within CT) produced comparable results [34]. The range of functional testing methods highlights the variability in how functional capacity is conceptualised and measured within CT studies.

Physical activity levels were examined in three trials using the International Physical Activity Questionnaire (IPAQ) [38,41], or pedometer-derived step counts [36]. Findings were mixed; some studies reported increases in physical activity following CT [41], whereas others observed minimal or no change in habitual physical activity or daily step counts [36,38]. These discrepancies underscore uncertainties in how physical activity behaviour responds to CT, as well as the limited number of studies investigating this outcome.

**Participation and quality of life.** Across the included studies, participants were adults with CAD enrolled in CR programmes, with key characteristics summarised in Table 2. Quality of life was evaluated using either the 36-Item Short Form Health Survey (SF-36) [41,42,44] or 12-Item Short Form Health Survey (SF-12) [40]. Several studies reported improvements in quality-of-life domains over time following CT or aerobic training, particularly in physical functioning [41,44]. However, between-group differences were inconsistent, with some studies reporting minimal changes in QoL outcomes [40,42]. This variability reflects the diversity of QoL instruments employed and the limited number of studies evaluating patient-reported outcomes within CT interventions.

### Adverse events

Of the 14 studies reviewed, nine reported no adverse events, whereas the remaining five did not specify adverse event outcomes. No hospitalisations or exercise-related complications were documented.

### Discussion

Cardiac rehabilitation has evolved from traditional aerobic-based models to more comprehensive programmes that integrate both aerobic and strength training to improve cardiovascular fitness and musculoskeletal health. This shift reflects growing recognition that exercise capacity, muscle strength, and functional performance are equally critical determinants of prognosis and quality of life in patients with CAD. Our discussion interprets patterns in how CT is designed, operationalised, and reported across the literature, and identify areas where greater standardisation and conceptual clarity are required.

### Patterns in CT delivery, study design, and CR phase

Across the included studies, designs were predominantly RCTs, with one single-group design. This methodological pattern reflects a strong emphasis on controlled comparison frameworks, despite considerable variation in the structure and

progression of CT protocols. CT was delivered across a wide range of settings, formats, and durations. Most interventions were implemented within hospital-based, supervised programmes, with fewer studies adopting community-based or hybrid models. This distribution indicates that CT research remains concentrated within structured clinical environments, with limited exploration of delivery models that may enhance accessibility or support longer-term engagement. Although supervised settings can improve safety and intervention fidelity, they may also restrict accessibility and reduce long-term adherence [45].

The CR phase also varied across studies. CT was applied in both Phase II and Phase III programmes, yet reporting often lacked clarity regarding the rationale for phase selection or the extent to which phase-specific goals informed protocol design. This inconsistency complicates interpretation of how CT is tailored to different stages of recovery. Programme duration ranged from 5 to 32 weeks, demonstrating substantial heterogeneity in the conceptualisation of training cycles. Training frequency also varied, from once weekly to five sessions per week. Such diversity suggests a lack of consensus on the optimal programme length or session volume required to achieve targeted outcomes, further illustrating the breadth of CT implementation approaches.

## Variability in aerobic and strength training prescription

Aerobic training varied substantially in modality (single-modality vs multimodal formats), duration, and intensity prescription. Studies employed CAT, AIT, and HIIT formats, with intensities anchored to a diverse range of physiological markers ($HR_{max}$, HRR, $W_{peak}$, VT power, $VO_2$peak, or RPE), consistent with contemporary CR guidelines [46–48]. Some studies reported progressive adjustments in workload, whereas others maintained fixed intensity throughout. These variations illustrate the breadth of approaches used to operationalise aerobic components within CT. Based on the available evidence, effective prescriptions may include 30–45 minutes of moderate-to-vigorous aerobic training, performed 3–5 times per week, at 60–80% $HR_{max}$ or 50–70% $VO_2$peak.

Strength training protocols also exhibited considerable heterogeneity but were generally aligned with clinical guidelines [48–50]. Prescriptions differed in sets, repetitions, intensity anchors (e.g., 1RM, 2RM, MVC, RPE), and the type of equipment used. Some programmes targeted major muscle groups of both upper and lower limbs using multi-exercise formats, whereas others focused predominantly on lower-limb musculature or single-exercise approaches. When applied, progressive overload was introduced through incremental changes in load, intensity, or training volume; however, this was not implemented consistently across studies. A pragmatic dosage for patients may consist of 2–3 non-consecutive sessions per week, comprising 2–3 sets of 10–15 repetitions at 40–70% 1RM, indicating current practices in the operationalisation of strength components within CT.

## Patterns across ICF outcome domains

The included studies assessed a range of outcome domains aligned with ICF domains. Exercise capacity, most frequently assessed using $VO_2$peak derived from cardiopulmonary or exercise stress testing was reported in six studies, although follow-up durations and reporting formats varied. Previous systematic reviews indicated that CT could enhance exercise capacity relative to aerobic training alone [51–53]. In the present scoping review, most CT trials included $VO_2$peak as a key outcome, but heterogeneity in protocol design and timing of assessments limits direct comparison of trajectories across studies.

Strength outcomes were evaluated with multiple instruments (e.g., 1RM/2RM tests, dynamometry, handgrip strength), resulting in variation in the muscle groups assessed and contraction types represented in the literature. Nevertheless, observed improvements in muscle strength following CT support its added value within CR, particularly given its contribution to functional independence and prognosis [13,53,54].

Functional performance and physical activity were measured less consistently across the included studies. Functional performance was assessed using tools such as the 6MWT, ISWT, and estimated METs, each capturing distinct

dimensions of activity tolerance. Only three studies evaluated habitual physical activity, using self-reported questionnaires or pedometer-recorded step counts. Reporting patterns varied: some studies observed increases in self-reported activity, whereas studies employing objective monitoring reported minimal or no measurable change. These discrepancies highlight variability in measurement approaches and the limited evidence available on how structured CT influences daily physical activity behaviours.

Participation-level outcomes, including HrQoL, were assessed using the SF-36 or SF-12 in a minority of trials. Findings varied across studies and across HrQoL domains, with limited long-term follow-up available. Some studies reported improvements in selected domains, while others documented minimal or no observable changes, particularly in emotional or mental health components [11].

### Gaps in the literature (PCC Framework)

**Population.** Most study samples predominantly comprised middle-aged to older males, with a notable under-representation of females (33%, n = 309). The inclusion of female participants in only two trials [35,44] highlights the limited evidence regarding sex differences in response to CT. This gap restricts understanding of whether females experience comparable physiological or functional adaptations to CT and represents an important priority for future research. Females entering CR generally exhibit lower baseline aerobic capacity and muscle strength than males [55,56]. Despite these lower starting values, improvements in peak $VO_2$ during Phase II CR are often less pronounced in females; one study reported mean gains of 0.3 L/min in females compared with 0.4 L/min in males [55]. Furthermore, many activities of daily living (ADLs), particularly among older females following a cardiac event, depend more heavily on muscular strength than endurance. Accordingly, strength training is vital for maintaining functional independence, enabling individuals to perform tasks such as stair climbing, carrying groceries, and housework [44].

**Concept.** Considerable heterogeneity was observed across CT protocols, with no consistent framework for prescribing frequency, intensity, duration, or progression of aerobic and strength components. A limited number of studies [13,34,42] examined how specific elements of CT, such as the combination of aerobic and strength modalities, the sequencing of exercises within a session, or variations in training intensity and frequency, may have contributed to the observed outcomes. These gaps highlight opportunities for future research to investigate how different CT configurations, including specific intensity–frequency combinations, influence both physiological adaptations and patient-reported outcomes. Gomes-Neto et al. [57] highlighted the need to clarify which specific training characteristics, such as intensity, frequency, and duration, optimise $VO_2$peak, HrQoL, and mortality outcomes. They also identified the need to determine whether energy expenditure, exercise type, or their combination best predicts improvements in $VO_2$peak in patients with CAD.

**Context.** Most CT protocols were delivered in hospital-based, supervised CR programmes. Community, home-based, hybrid, and telehealth-supported models were rarely investigated, despite their potential relevance for accessibility and long-term maintenance. Differences in CR phase (Phase II vs Phase III) and inconsistent reporting of phase rationale further limit comparability. Long-term follow-up beyond the immediate programme period was uncommon, providing little insight into the durability of changes in capacity, strength, activity, or HrQoL once supervision ends.

### Implications for future research

Future research would benefit from more consistent reporting of CT prescription, including justification for selected intensities, modalities, progression strategies, and total training dose (e.g., energy expenditure and combined aerobic–strength load). Expanding CT investigations into community, hybrid, and telehealth CR models may help address contextual barriers to participation and support more scalable delivery. The incorporation of systematically collected patient-reported outcomes could enhance understanding of how individuals integrate CT into daily life and how changes in capacity relate to long-term physical activity and quality of life. Evidence on gender-specific responses to CT remains limited and warrants further investigation. Systematic evaluation of varying combinations and doses of aerobic and strength components,

supported by more diverse samples, sex-stratified analyses, and longer-term follow-up, may strengthen the refinement of CT programmes and guide future recommendations without assuming a single "optimal" approach.

## Conclusion

This scoping review mapped how CT has been implemented within CR, highlighting substantial variation in programme delivery, training prescription, and outcome measurement across studies. While exercise capacity, muscle strength, and functional performance were frequently assessed, physical activity and participation-level outcomes were evaluated less consistently. The heterogeneity observed across study designs, CR phases, settings, and assessment tools underscores the need for more standardised reporting frameworks and clearer conceptualisation of concurrent training components. Future research would benefit from broader representation of patient populations, greater inclusion of patient-reported outcomes, and the examination of CT within community, hybrid, and telehealth models to support more scalable and contextually adaptable approaches to CR programmes.

## Supporting information

**S1 File. Full Electronic Search Strategy (October 2023; updated Dec 2025).** The table presents the complete search strings, limits, filters, and retrieval dates applied in PubMed (MEDLINE), Scopus, Web of Science Core Collection, and the Cochrane Library (CENTRAL). The search strategy incorporated terms related to aerobic training, resistance training, cardiac rehabilitation, and coronary artery disease. Limits were applied according to each database's functions, including study type, English language, adult populations, and publication years (2000–2025), with preprints excluded when applicable. The final number of records retrieved from each database is provided.
(PDF)

**S2 File. PRISMA-ScR-Fillable-Checklist used for the scoping review.** This checklist demonstrates adherence to the Preferred Reporting Items for Systematic Reviews and Meta-Analyses extension for Scoping Reviews (PRISMA-ScR). Each item aligns with the recommended reporting domains, such as title, abstract, introduction, methods, results, discussion, and funding information. The completed form specifies the location of each required element within the manuscript, thereby promoting transparency and compliance with established methodological and reporting standards.
(PDF)

## Acknowledgments

The authors acknowledge University of Malaya (UM) and Universiti Teknologi MARA (UiTM) and libraries for their assistance and access to electronic databases that facilitated this research.

## Author contributions

**Conceptualization:** Aliff Latir.

**Data curation:** Aliff Latir, Eliza Hafiz.

**Writing – original draft:** Aliff Latir.

**Writing – review & editing:** Aliff Latir, Eliza Hafiz, Anwar Suhaimi.

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
