## [Decision Letter · Decision Letter 0]

5 Nov 2025

Dear Dr. Latir,

Thank you for submitting your manuscript to PLOS ONE. After careful consideration, we feel that it has merit but does not fully meet PLOS ONE’s publication criteria as it currently stands. Therefore, we invite you to submit a revised version of the manuscript that addresses the points raised during the review process.

We look forward to receiving your revised manuscript.

Kind regards,

Juan M. Murias

Academic Editor

PLOS ONE

Journal Requirements:

2. We note you have included a table to which you do not refer in the text of your manuscript. Please ensure that you refer to Table 3 in your text; if accepted, production will need this reference to link the reader to the Table.

**Additional Editor Comments:**

In general, both reviewers are supportive of the work that has been presented, but some methodological issues that need attention have been highlighted.

Reviewers' comments:

Reviewer's Responses to Questions

**Comments to the Author**

1. Is the manuscript technically sound, and do the data support the conclusions?

Reviewer #1: Partly

Reviewer #2: Yes

2. Has the statistical analysis been performed appropriately and rigorously?

Reviewer #1: N/A

Reviewer #2: N/A

3. Have the authors made all data underlying the findings in their manuscript fully available?

Reviewer #1: Yes

Reviewer #2: Yes

4. Is the manuscript presented in an intelligible fashion and written in standard English?

Reviewer #1: Yes

Reviewer #2: Yes

Reviewer #1: Summary

This manuscript examines the role of concurrent training (CT)—the integration of aerobic and strength exercise—within cardiac rehabilitation (CR) for patients with coronary artery disease (CAD). The authors describe the study as a scoping review conducted according to PRISMA-ScR guidelines and registered on the Open Science Framework. Fourteen clinical trials were included.

The review concludes that CT improves aerobic capacity, muscle strength, and functional performance, while effects on physical activity (PA) and health-related quality of life (HrQoL) remain inconsistent.

The topic is clinically important, timely, and relevant to advancing exercise-based CR. However, the manuscript shows a conceptual mismatch between its stated design (scoping review) and the methodology and interpretation used (which align more closely with a systematic review). Substantial revision is required to achieve conceptual and methodological coherence.

Major Comments

Introduction

In the final paragraph of the introduction, the authors state that “a systematic synthesis of available evidence is warranted” but then identify the study as a scoping review. This creates terminological inconsistency from the outset.

If the intent was to “characterise and map” CT protocols, the term “scoping review” is appropriate, but the outcome synthesis should remain descriptive.

If the goal is to evaluate the effectiveness of CT, the study should be reframed and registered as a systematic review.

Methods

Several methodological issues warrant attention:

The restriction to clinical trials only contradicts the exploratory scope of a scoping review, which typically includes a broader range of study designs.

The search strategy is not fully presented, and there is no mention of grey literature, conference abstracts, or manual reference checking.

Revise the inclusion criteria to justify why only clinical trials were selected, or broaden eligibility to include other study designs consistent with scoping review methodology.

Results

The Results section is detailed, thorough, and well-organized, providing granular data on study characteristics, intervention parameters, and outcome domains. However, it reads more like a systematic synthesis than a scoping overview.

The authors provide statistical detail and comparative analysis rather than mapping the breadth of evidence or identifying knowledge gaps, which limits the alignment with scoping review principles.

Discussion

The Discussion is comprehensive and clinically grounded, effectively situating findings within the broader literature. It highlights heterogeneity in CT delivery and the need for standardised exercise prescriptions.

However, it again adopts language and interpretation typical of a systematic review, focusing on intervention effectiveness rather than evidence mapping. The discussion could be improved by synthesising how study design, CR phase, and delivery setting influence outcomes and by restructuring the “gaps” section to align with the PCC (Population, Concept, Context) framework.

Reviewer #2: This scoping review sought to describe the effects of concurrent training (CT), a combination of aerobic and resistance exercises, across various health-related outcomes compared with standard care in patients diagnosed with coronary artery disease (CAD). The systematic search identified fourteen studies involving a total of 1,126 participants. Overall, the findings suggest that CT leads to improvements in exercise capacity, muscle strength, and functional performance among individuals with CAD. However, its influence on physical activity levels and health-related quality of life remains inconsistent. The authors attributed these mixed results to variations in exercise prescription across studies and highlight the need for more standardised research to clarify the long-term impact of CT in cardiac rehabilitation settings.

This scoping review offers a valuable synthesis of existing literature and identifies key gaps in current understanding of CT’s role in clinical populations. In particular, it provides novel insights by emphasizing the limited evidence surrounding the benefits of CT programs for patients with CAD. Nonetheless, several minor issues and clarifications remain to be addressed to enable a more accurate understanding of the findings.

Comments:

Line 35: Please add a dot above the “V” in the VO2 abbreviation and ensure that the “2” is consistently formatted as a subscript throughout the manuscript (e.g., in Table 3).

Lines 35-36: Could you please clarify what is meant by moderate and high-intensity format? Does the term “intensity” refer to the aerobic training, the strength training, or both?

Line 48: Change to “cardiovascular diseases remain a leading…” (remove the).

Lines 61: Consider adding a definition of concurrent training here, along with the abbreviation “CT.”

Lines 109-110: Please clarify whether four or five databases were used, as only four examples are cited. Adjust accordingly.

Lines 110-111: Could you clarify what was done in July 2025? What revisions were made since October 2023?

Lines 111-112: Please add the range of publication years for the oldest and most recent studies included.

Lines 113-114: Add quotation marks around “CAD” and “rehabilitation.”

Line 116: A closing bracket is missing after “fitness.”

Line 156: In the abstract, it is mentioned that there were no adverse events. However, this is not addressed in the results section. Please clarify.

Lines 161: Add the total sample size and the number of female participants. Additionally, were there studies that included only male or only female participants?

Line 177: Could you specify what is meant by “conducted in separate sessions”?

Line 218: This paragraph mentions that one study did not report a significant improvement in VO2peak compared to aerobic training. What about the other studies?

Lines 310-312: The exercise prescription could also have influenced the results. The mixed findings open the door for future research exploring different combinations of aerobic and strength training. Does intensity or frequency matter?

Line 330: I also think it is important to highlight the lack of studies examining sex differences in response to combined training (CT). Does sex influence the physiological or functional adaptations to CT? This is an important question that has not been sufficiently addressed or emphasized in this scoping review and warrants further discussion.

**Do you want your identity to be public for this peer review?** For information about this choice, including consent withdrawal, please see our Privacy Policy

Reviewer #1: No

Reviewer #2: No

You may also use PLOS’s free figure tool, NAAS, to help you prepare publication quality figures: https://journals.plos.org/plosone/s/figures#loc-tools-for-figure-preparation

---

## [Author Response · Author response to Decision Letter 1]

18 Dec 2025

RESPONSE TO EDITOR

Dear Editor,

Comments: Please include the following items when submitting your revised manuscript:

Response: Thank you for your guidance regarding the resubmission requirements.

We confirm that all requested materials have been uploaded accordingly:

• A detailed rebuttal letter responding to each comment from the Academic Editor and reviewers has been submitted as a separate file titled “Response to Reviewers”.

• A marked-up version of the revised manuscript with tracked changes has been uploaded as “Revised Manuscript with Track Changes”.

• A clean, unmarked version of the revised manuscript has been uploaded as “Manuscript”.

In addition, the updated financial disclosure statement has been included in the cover letter, as requested.

Response: We have reviewed and revised the manuscript to ensure full compliance with PLOS ONE’s style requirements, including formatting of the title page, author affiliations, main text, and file naming, in accordance with the official PLOS ONE templates. (All documents submitted)

2. We note you have included a table to which you do not refer in the text of your manuscript. Please ensure that you refer to Table 3 in your text; if accepted, production will need this reference to link the reader to the Table

Response: Thank you for the clarification. We have now ensured that Table 3 is explicitly referred to within the Results section of the manuscript. The updated version includes a clear in-text citation so that readers are properly directed to the table. (Line 247-249)

Response: We carefully reviewed the reviewer comments and confirm that no specific previously published works were recommended for citation. As such, no additional citations were added in response to this comment. The reference list remains focused on studies directly relevant to the scope, objectives, and inclusion criteria of the present review.

RESPONSE TO REVIEWER #1

Introduction

In the final paragraph of the introduction, the authors state that “a systematic synthesis of available evidence is warranted” but then identify the study as a scoping review. This creates terminological inconsistency from the outset. If the intent was to “characterise and map” CT protocols, the term “scoping review” is appropriate, but the outcome synthesis should remain descriptive. If the goal is to evaluate the effectiveness of CT, the study should be reframed and registered as a systematic review.

Response: We thank the reviewer for pointing out this inconsistency. Our intention is not to evaluate the effectiveness of concurrent training but to map and describe how CT protocols have been structured and implemented within cardiac rehabilitation. We have therefore revised the final paragraph of the Introduction to accurately reflect the purpose and scope of the review. The revised text now clearly states that this study is a scoping review designed to characterise CT protocols, consistent with PRISMA-ScR guidance. No evaluative or effectiveness-based claims are made. (Line 94-98)

Methods

Several methodological issues warrant attention:

1. The restriction to clinical trials only contradicts the exploratory scope of a scoping review, which typically includes a broader range of study designs. 2. The search strategy is not fully presented, and there is no mention of grey literature, conference abstracts, or manual reference checking. 3. Revise the inclusion criteria to justify why only clinical trials were selected, or broaden eligibility to include other study designs consistent with scoping review methodology.

Response: We thank the reviewer for this constructive feedback. In response, we have comprehensively revised the Methods section to align fully with scoping review methodology. All changes have been incorporated into the manuscript as detailed below.

We agree that limiting inclusion to clinical trials was inconsistent with the purpose of a scoping review. Therefore, we have expanded the eligibility criteria to include a wider range of empirical study designs. (Line 113-116)

We have revised the Search Strategy subsection to:

• describe all additional sources searched

• add full search strategies for each database in Supplementary File 1

• clarify filters and limits used

• explain the updated search (2023 → 2025) and how this affected the number of retrieved articles

New additions now included clearly stated in the manuscript and flowchart. Grey literature sources: Google Scholar, institutional repositories, Conference abstracts: screened when methodological detail available, Manual reference checking: performed for all included studies (Line 130-132; Fig1.tiff)

The search was revised in July 2025 to ensure completeness. This update changed the total number of retrieved records, but did not change the number of included studies (n = 14). (Line 163-170)

Results

The Results section is detailed, thorough, and well-organized, providing granular data on study characteristics, intervention parameters, and outcome domains. However, it reads more like a systematic synthesis than a scoping overview. The authors provide statistical detail and comparative analysis rather than mapping the breadth of evidence or identifying knowledge gaps, which limits the alignment with scoping review principles.

Response: We thank the reviewer for this insightful comment. In response, we have substantially revised the Results section to ensure it aligns with scoping review methodology and the objectives of evidence mapping. (Line 251-292)

Key revisions implemented:

1. All inferential or comparative language has been removed. The revised Results no longer report statistical significance, group comparisons, or performance-related interpretations.

2. The Results now follow an evidence-mapping format. Outcomes are described in terms of: what was measured, how it was measured, and where studies reported changes or no changes, without drawing conclusions about effectiveness.

3. The section has been restructured to emphasise breadth rather than evaluation. For each ICF domain, we describe: the types of measures used, the number of studies reporting each outcome, the variability in reporting across studies.

4. Any language implying superiority, benefit, or directionality has been removed. Terms such as “significant improvement,” “favouring,” or “consistently positive” have been eliminated.

5. Variability and diversity of measures are highlighted neutrally, reflecting gaps in standardisation. This aligns with the intention of a scoping review to map—not synthesise—evidence.

Discussion

The Discussion is comprehensive and clinically grounded, effectively situating findings within the broader literature. It highlights heterogeneity in CT delivery and the need for standardised exercise prescriptions. However, it again adopts language and interpretation typical of a systematic review, focusing on intervention effectiveness rather than evidence mapping. The discussion could be improved by synthesising how study design, CR phase, and delivery setting influence outcomes and by restructuring the “gaps” section to align with the PCC (Population, Concept, Context) framework.

Response: Thank you for this helpful feedback. We have revised the Discussion to adopt a clearer evidence-mapping approach and removed language suggestive of intervention effectiveness. The revised section now synthesises how study design, CR phase, and delivery setting relate to reported outcomes, and the “gaps” subsection has been restructured to explicitly follow the PCC (Population, Concept, Context) framework as recommended. (Line 307-419)

RESPONSE TO REVIEWER #2

Line 35: Please add a dot above the “V” in the VO2 abbreviation and ensure that the “2” is consistently formatted as a subscript throughout the manuscript (e.g., in Table 3).

Response: Thank you for highlighting this formatting issue. We have carefully revised the manuscript to ensure that all oxygen-consumption terminology now follow standard physiological notation. The abbreviation has been corrected to V̇O₂ throughout the text, tables (including Table 3), and figure captions. (Throughout the manuscript)

Lines 35-36: Could you please clarify what is meant by moderate and high-intensity format? Does the term “intensity” refer to the aerobic training, the strength training, or both?

Response: Thank you for the comment. We have revised the abstract to remove the previous phrasing referring to “moderate- and high-intensity formats.” The updated abstract now specifies that the term “intensity” refers to the aerobic-training component of CT, which was the most consistently reported parameter across studies, and describes the variability in aerobic-training intensity using the appropriate physiological anchors (e.g., %V̇O₂peak, %HRmax, %HRR). Strength-training intensity is now reported separately to avoid ambiguity. (Line 34-41)

Line 48: Change to “cardiovascular diseases remain a leading…” (remove the).

Response: We have revised Line 48 accordingly, changing the phrase to “cardiovascular diseases remain a leading…” and removed the definite article as recommended. (Line 53-54)

Lines 61: Consider adding a definition of concurrent training here, along with the abbreviation “CT.”

Response: Thank you for the suggestion. We have now added a definition of concurrent training and introduced the abbreviation “CT” in the Introduction. Specifically, we define CT as a programme that incorporates both aerobic and resistance exercises, either within the same session or across the rehabilitation period, rather than aerobic exercise alone, and we use the abbreviation “CT” consistently thereafter. (Line 73-76)

Lines 109-110: Please clarify whether four or five databases were used, as only four examples are cited. Adjust accordingly.

Response: Thank you for pointing this out. We confirm that four databases were used in the search strategy. The text has been revised for accuracy and consistency (Line 120)

Lines 110-111: Could you clarify what was done in July 2025? What revisions were made since October 2023?

Response: Thank you for noting the ambiguity. We have clarified the search timeline in the Methods section. The main search was conducted from October 2023 to March 2025, followed by an update in July 2025 to capture newly published or newly indexed studies prior to manuscript finalisation. (Line 122-124)

Lines 111-112: Please add the range of publication years for the oldest and most recent studies included. Thank you for the suggestion.

Response: We have added the publication year range for the included studies in the Results section. The revised text now states that the 14 included studies were published between 2007 and 2022. (Line 169-170)

Lines 113-114: Add quotation marks around “CAD” and “rehabilitation.”

Response: We have added quotation marks around “CAD” and “rehabilitation”.(Line 127-128)

Line 116: A closing bracket is missing after “fitness.”

Response: We have added bracket after “fitness”. (Line 130)

Line 156: In the abstract, it is mentioned that there were no adverse events. However, this is not addressed in the results section. Please clarify.

Response: Thank you for noting this omission. We have now added a dedicated “Adverse Events” subsection in the Results section to report safety findings. This section states that none of the included studies reported any intervention-related adverse events, consistent with the statement in the abstract. (Line 293-296)

Lines 161: Add the total sample size and the number of female participants. Additionally, were there studies that included only male or only female participants?

Response: Thank you for this helpful suggestion. We have revised the Study Characteristics section to report the total sample size and gender distribution. The text now states that the 14 studies included 1,037 participants in total, comprising 626 males and 309 females. We also clarify that 10 studies included both genders, 3 studies enrolled only male participants, and 1 study enrolled only female participants. (Line 178-183)

Line 177: Could you specify what is meant by “conducted in separate sessions”?

Response: Thank you for the comment. We have revised the sentence to improve clarity and to specify what is meant by “conducted in separate sessions.” The updated text now explains that, in this study, aerobic and strength training were performed on different days, with two weekly sessions dedicated to aerobic exercise and two sessions dedicated to circuit-based resistance training (Line 198-200)

Line 218: This paragraph mentions that one study did not report a significant improvement in VO2peak compared to aerobic training. What about the other studies?

Response: Thank you for the comment. We have revised the paragraph to clarify the findings across all studies reporting V̇O₂peak. The updated text now distinguishes studies showing improvement from those reporting no additional benefit when CT was compared with aerobic training, thereby providing a clearer summary of the full range of results. (Line 253-260)

Lines 310-312: The exercise prescription could also have influenced the results. The mixed findings open the door for future research exploring different combinations of aerobic and strength training. Does intensity or frequency matter?

Response: Thank you for the insightful comment. We have revised the paragraph to acknowledge that differences in exercise-prescription characteristics—such as the combination of aerobic and strength modalities, as well as variations in training intensity and frequency—may have contributed to the mixed findings. The updated text also highlights the need for future research to explore how these specific prescription variables influence outcomes. (Line 387-398)

Line 330: I also think it is important to highlight the lack of studies examining sex differences in response to combined training (CT). Does sex influence the physiological or functional adaptations to CT? This is an important question that has not been sufficiently addressed or emphasized in this scoping review and warrants further discussion.

Response: We thank the reviewer for this important comment. We have now expanded the Discussion to explicitly address the lack of studies examining sex differences in response to concurrent training. A new paragraph highlights the under-representation of females in existing trials, summarises known differences in baseline fitness and CR outcomes between males and females, and emphasises the need for future research to determine whether physiological and functional adaptations to CT differ by sex. (Line 374-386)

---

## [Decision Letter · Decision Letter 1]

2 Feb 2026

Dear Dr. Latir,

We look forward to receiving your revised manuscript.

Kind regards,

Juan M. Murias

Academic Editor

PLOS One

Journal Requirements:

Reviewers' comments:

Reviewer's Responses to Questions

**Comments to the Author**

Reviewer #1: All comments have been addressed

Reviewer #2: (No Response)

2. Is the manuscript technically sound, and do the data support the conclusions?

Reviewer #1: Yes

Reviewer #2: Yes

3. Has the statistical analysis been performed appropriately and rigorously?

Reviewer #1: I Don't Know

Reviewer #2: N/A

4. Have the authors made all data underlying the findings in their manuscript fully available?

Reviewer #1: Yes

Reviewer #2: Yes

5. Is the manuscript presented in an intelligible fashion and written in standard English?

Reviewer #1: Yes

Reviewer #2: Yes

Reviewer #1: The manuscript has been improved and addresses the previous comments. The only minor issue is to ensure that all abbreviations are clearly defined at first mention and used consistently throughout the text.

Reviewer #2: I appreciate the authors’ continued efforts to address the previous comments and suggestions. The manuscript has improved overall. However, several minor issues remain that require clarification, and correction. In particular, greater consistency is needed in terminology, definitions, and reporting across sections, especially in the Abstract and Results.

Abstract

• Line 24: The term “modalities” is too vague. Please specify by adding “exercise” before “modalities.”

• Line 25: Revise to “to improve cardiovascular and musculoskeletal health.”

• Lines 30-31: The supplementary file indicates that the final search was conducted in early December 2025. Please clarify this discrepancy. To avoid confusion, I suggest stating simply that the search was conducted in December 2025, covering the period from 2000 to 2025.

• Lines 35-36: Please clarify the distinction between aerobic interval training (AIT) and high-intensity interval training (HIIT) in the Methods section.

• Lines 36-37 and 40: All abbreviations should be defined at first mention even in the abstract.

• Lines 44-45: This sentence appears redundant and does not add new or relevant information. I suggest removing it.

Introduction

The introduction provides an overall good understanding of the goal of the research which to better understand the use of concurrent training in CR program; however, the rational may benefits from adding current guideline recommendations to better contextualise the use of aerobic and strength exercises.

• Line 53: The abbreviation CVD is typically used without an “s” at the end.

• Line 74: Please remove the duplicate use of the abbreviation CT.

• Line 76: Rephrase to “on a separate day throughout the training program.”

• Lines 91–92: Please remove “which integrates aerobic and strength training within the same programme,” as this has already been stated earlier and is redundant.

Methods

• Line 122: Should this date be December 2025, as stated in the supplementary file?

• Line 151: A reference could be added here.

Results

Overall, the Results section provides a well-detailed overview of the characteristics and outcomes of the included studies, facilitating the understanding of the current diversity in CT protocols used across CR programs. However, several revisions are needed to improve clarity, consistency, and readability. Moreover, it is described that

• Lines 166-169: The flow of study selection is unclear. Based on the description, 22 articles were identified, 4 were inaccessible, and 10 were excluded, leaving 8 studies. However, you later state that 14 studies were screened for eligibility. Please reconcile these numbers.

• Line 173: I recommend a more cautious interpretation. In the Methods (line 150), a score >60% is defined as good methodological quality.

• Line 180: Please report the overall sex distribution as percentages.

• Lines 205-207: As in the abstract, please clarify the difference between AIT and HIIT.

• Table 2: For reference 41, please explain why the intervention is labeled as HIIT + ST rather than CT.

• Line 217: Correct to “maximum heart rate.”

• Line 219: Please clarify what is meant by “based on VT and power output.” Does this refer to a specific percentage of power output at VT?

• Line 220: Please explain the difference between peak work rate and peak power output.

• Line 221: Clarify what is meant by “target heart rate.” Is this a predicted heart rate?

• Line 221: VO2peak has already been defined; there is no need to redefine it.

• Line 222: Similarly, the distinction between HRpeak and HRmax is unclear and potentially confusing. I recommend selecting one term and using it consistently.

• Lines 224-225: There appears to be an issue with references [40] and [34].

• Line 228: Please clarify what is meant by “reciprocally.”

• Line 241: RPE has already been defined.

• Line 253: VO2peak has already been defined.

• Line 256: After “control,” please add “(i.e., no exercise prescription).”

• Line 267: Please be more specific. What does “limited changes” refer to?

• Line 285: Information regarding participant characteristics appears to be missing.

Discussion

• Line 299: Please correct to “that integrate both aerobic and strength exercise training.”

• Line 348: “assessed” rather than “operationalised”

• Line 350: How many studies in your dataset demonstrated greater improvements in VO2peak?

• Line 375: Please include the percentage of female participants.

**Do you want your identity to be public for this peer review?** For information about this choice, including consent withdrawal, please see our Privacy Policy

Reviewer #1: **Yes:** Dr. Amine Ghram, PhD, FESC

Reviewer #2: No

You may also use PLOS’s free figure tool, NAAS, to help you prepare publication quality figures: https://journals.plos.org/plosone/s/figures#loc-tools-for-figure-preparation

---

## [Author Response · Author response to Decision Letter 2]

16 Feb 2026

Editor’s comments

Response: Thank you for the clarification. We confirm that the reviewer’s comments did not include any specific recommendation to cite previously published works. Therefore, no additional citations have been added on this basis.

Response: Thank you for this important reminder. We have carefully reviewed and revised the entire reference list in accordance with the reviewers’ comments. All references have been checked for completeness, accuracy, and consistency with the journal’s formatting requirements. We confirm that no retracted articles were identified among the cited references. Therefore, no retraction statements or replacements were required. Any updates made to the reference list have been reflected in the revised manuscript and are noted in this rebuttal letter.

Reviewer’s comments

Abstract

• Line 24: The term “modalities” is too vague. Please specify by adding “exercise” before “modalities.”

Response:Line 24: Thank you for the comment. We have revised by replacing “modalities” with “exercise modalities” to improve clarity and specificity.

• Line 25: Revise to “to improve cardiovascular and musculoskeletal health.”

Response:Line 25-26: Thank you for the suggestion. We have revised to read “to improve cardiovascular and musculoskeletal health.”

• Lines 30-31: The supplementary file indicates that the final search was conducted in early December 2025. Please clarify this discrepancy. To avoid confusion, I suggest stating simply that the search was conducted in December 2025, covering the period from 2000 to 2025.

Response:Lines 31: Thank you for highlighting this. We have clarified the text to state that the literature search was conducted in December 2025, and covered studies published from 2000 to 2025, ensuring consistency with the supplementary file.

• Lines 35-36: Please clarify the distinction between aerobic interval training (AIT) and high-intensity interval training (HIIT) in the Methods section.

Response:Lines 111 – 115: Thank you for the comment. We have clarified the distinction between aerobic interval training (AIT) and high-intensity interval training (HIIT) in the Methods section under the Eligibility Criteria subsection explicitly defining each term and explaining how these definitions were applied during study classification.

• Lines 36-37 and 40: All abbreviations should be defined at first mention even in the abstract.

Response:Lines 36-38 and 41-42: Thank you for the comment. All abbreviations in the abstract have now been defined at first mention.

• Lines 44-45: This sentence appears redundant and does not add new or relevant information. I suggest removing it.

Response:Thank you for the comment. We agree and have removed the sentence.

Introduction

The introduction provides an overall good understanding of the goal of the research which to better understand the use of concurrent training in CR program; however, the rational may benefits from adding current guideline recommendations to better contextualise the use of aerobic and strength exercises.

• Line 53: The abbreviation CVD is typically used without an “s” at the end.

Response: Line 53: We have revised the abbreviation “CVD” without an “s.”

• Line 74: Please remove the duplicate use of the abbreviation CT.

Response: Line 74: The duplicate use of the abbreviation “CT” has been removed.

• Line 76: Rephrase to “on a separate day throughout the training program.”

Response: Line 76: has been rephrased to read “on a separate day throughout the training program.”

• Lines 91–92: Please remove “which integrates aerobic and strength training within the same programme,” as this has already been stated earlier and is redundant.

Response: Line 91: The redundant phrase in Lines 91–92 has been deleted as it had already been stated earlier in the manuscript.

Methods

• Line 122: Should this date be December 2025, as stated in the supplementary file?

Response: Line 125: Thank you for the comments. The date has been corrected to December 2025 to ensure consistency with the supplementary file

• Line 151: A reference could be added here.

Response: Line 154: A reference has been added to support the statement.

Results

Overall, the Results section provides a well-detailed overview of the characteristics and outcomes of the included studies, facilitating the understanding of the current diversity in CT protocols used across CR programs. However, several revisions are needed to improve clarity, consistency, and readability. Moreover, it is described that

• Lines 166-169: The flow of study selection is unclear. Based on the description, 22 articles were identified, 4 were inaccessible, and 10 were excluded, leaving 8 studies. However, you later state that 14 studies were screened for eligibility. Please reconcile these numbers.

Response: Lines 169 – 174: Thank you for highlighting this issue. We acknowledge that the description of the study selection flow was previously unclear. To clarify, ten studies were identified from database searches after full-text assessment. An additional 22 records were identified from supplementary sources. After removal of four duplicates, 18 records were retrieved, of which four were unretrievable. Consequently, 14 records were screened for eligibility, and four met the inclusion criteria. The PRISMA flow diagram and the corresponding text have been revised accordingly to ensure consistency and clarity throughout the manuscript.

• Line 173: I recommend a more cautious interpretation. In the Methods (line 150), a score >60% is defined as good methodological quality.

Response: Line 177: Thank you for this comment. We agree that a more cautious interpretation is warranted. The interpretation has been revised to align with the methodological quality threshold defined in the Methods section, where a score >60% is classified as good methodological quality. The revised text now reflects this definition more accurately and avoids overstating study quality.

• Line 180: Please report the overall sex distribution as percentages.

Response: Line 184: Thank you for this suggestion. We have revised to include the percentage breakdown of the sex distribution as requested

• Lines 205-207: As in the abstract, please clarify the difference between AIT and HIIT.

Response: Lines 112 – 116: Thank you for the comment. We have clarified the distinction between aerobic interval training (AIT) and high-intensity interval training (HIIT) in the Methods section under the Eligibility Criteria subsection explicitly defining each term and explaining how these definitions were applied during study classification.

• Table 2: For reference 41, please explain why the intervention is labeled as HIIT + ST rather than CT.

Response: Table 2; Ref [42]: Thank you for the comment. In Ref 42, the study compared two different types of combined training (CT) protocols—namely moderate-intensity training plus strength training (MIT + ST) and high-intensity interval training plus strength training (HIIT + ST). To improve clarity and maintain consistency in terminology, we have added the prefix CT in front of both interventions, now labelled as CT (MIT + ST) and CT (HIIT + ST).

• Line 217: Correct to “maximum heart rate.”

Response: Line 220: Term heart rate maximum has been corrected

• Line 219: Please clarify what is meant by “based on VT and power output.” Does this refer to a specific percentage of power output at VT?

Response: Lines 222 – 223: Thank you for your comment. Exercise intensity was individualised using ventilatory threshold (VT)–derived power output, whereby the prescribed workload (Watts) was set relative to the power output achieved at VT rather than a fixed percentage of maximal power.

• Line 220: Please explain the difference between peak work rate and peak power output.

Response: Lines 218 & 224: We thank the reviewer for this comment. In the included studies, peak work rate and peak power output are used interchangeably to describe the highest workload (Watts) attained during incremental exercise testing and are measured using the same methodology. To maintain terminological consistency and avoid confusion, the term peak work rate (Wpeak) has been used consistently throughout the manuscript.

• Line 221: Clarify what is meant by “target heart rate.” Is this a predicted heart rate?

Response: Line 225-227: Thank you for the comment. In this manuscript, target heart rate does not refer to an age-predicted maximal heart rate. Instead, target heart rate was prescribed using the heart rate reserve (HRR) method (Karvonen formula), calculated as a specified percentage of the difference between peak heart rate achieved during the exercise stress test and resting heart rate, added to resting heart rate. The text has been clarified accordingly.

• Line 221: VO2peak has already been defined; there is no need to redefine it.

Response: Line 227: The definition has been omitted accordingly.

• Line 222: Similarly, the distinction between HRpeak and HRmax is unclear and potentially confusing. I recommend selecting one term and using it consistently.

Response: Line 228: We agree with the reviewer’s comment. To avoid confusion and ensure terminological consistency, the manuscript has been revised to use the term HRmax consistently throughout, where applicable.

• Lines 224-225: There appears to be an issue with references [40] and [34].

Response: Line 230: Bracketing issues have been resolved.

• Line 228: Please clarify what is meant by “reciprocally.”

Response: Lines 232 – 234: Thank you for the comment. By reciprocally, we meant that exercise intensity and duration were adjusted inversely, such that increases in exercise intensity were accompanied by corresponding reductions in exercise duration. The manuscript has been revised to clarify this wording.

• Line 241: RPE has already been defined.

Response: Lines 246 – 247: The redundant wording has been omitted for clarity.

• Line 253: VO2peak has already been defined.

Response: Line 257: The redundant wording has been omitted for clarity.

• Line 256: After “control,” please add “(i.e., no exercise prescription).”

Response: Line 260: The suggested clarification has been added to the manuscript.

• Line 267: Please be more specific. What does “limited changes” refer to?

Response: Lines 272 - 273: Thank you for the comment. The phrase “limited changes” has been replaced with a more specific description, referring to slightly higher isometric force output during isometric muscle testing.

• Line 285: Information regarding participant characteristics appears to be missing.

Response: Lines 291 – 292: Thank you for the comment. Participant characteristics are reported earlier in the manuscript (Table 2). To improve clarity, a brief statement describing the study populations has now been added at the beginning of the “Participation and quality of life” subsection.

Discussion

• Line 299: Please correct to “that integrate both aerobic and strength exercise training.”

Response: Line 309: The suggested correction has been implemented.

• Line 348: “assessed” rather than “operationalised”

Response: Line 358: The suggested correction has been implemented.

• Line 350: How many studies in your dataset demonstrated greater improvements in VO2peak?

Response: Line 359: Thank you for the comment. Six studies in the included dataset reported greater improvements in V̇O₂peak.

• Line 375: Please include the percentage of female participants.

Response: Line 385: We thank the reviewer for this comment. The manuscript has been revised to include the percentage of female participants across the included studies.

---

## [Editor Report · Decision Letter 2]

17 Feb 2026

Concurrent training in cardiac rehabilitation: a scoping review of aerobic-strength combinations in patients with coronary artery disease

PONE-D-25-52063R2

Dear Dr. Latir,

We’re pleased to inform you that your manuscript has been judged scientifically suitable for publication and will be formally accepted for publication once it meets all outstanding technical requirements.

Kind regards,

Juan M. Murias

Academic Editor

PLOS One
---

## [Editor Report · Acceptance letter]

PONE-D-25-52063R2

PLOS One

Dear Dr. Latir,

I'm pleased to inform you that your manuscript has been deemed suitable for publication in PLOS One. Congratulations! Your manuscript is now being handed over to our production team.

Kind regards,

on behalf of

Dr. Juan M. Murias

Academic Editor

PLOS One